# The Antibacterial Effects of Quaternary Ammonium Salts in the Simulated Presence of Inhibitors in Root Canals: A Preliminary In-Vitro Study

**Sanjay Kumar Tiwari [1,2,†]** , **Suping Wang [3,†]** , **Yannan Huang [1,2,†]** , **Xuedong Zhou [1,2]** , **Hockin H. K. Xu [4]** , **Biao Ren [1]** , **Xian Peng [1]** , **Michael D. Weir [4]** , **Mingyun Li [1,*]** and **Lei Cheng [1,2,*]**

[1] State Key Laboratory of Oral Diseases, West China school of Stomatology, Sichuan University, Chengdu 610041, China; sanjaytiwari_scu@hotmail.com (S.K.T.); susynan@163.com (Y.H.); zhouxd@scu.edu.cn (X.Z.); renbiao@scu.edu.cn (B.R.); pengx@scu.edu.cn (X.P.)

[2] Department of Operative Dentistry and Endodontics, West China school of Stomatology, Sichuan University, Chengdu 610041, China

[3] Stomatology Center, The First Affiliated Clinical Medical College of Zhengzhou University, Zhengzhou 450052, Henan, China; wangsupingdent@163.com

[4] Department of Endodontics, Periodontics and Prosthodontics, University of Maryland Dental School, Baltimore, MD 21201, USA; hxu@umaryland.edu (H.H.K.X.); Michael.weir@umaryland.edu (M.D.W.)

[*] Correspondence: limingyun@scu.edu.cn (M.L.); chenglei@scu.edu.cn (L.C.)

[†] These authors contributed equally to the article and share first authorship.

**Abstract:** To investigate the antibacterial effects of two newly developed quaternary ammonium salts (QAMs)-dimethylaminododecyl methacrylate (DMADDM) and dimethylaminohexadecyl methacrylate (DMAHDM), in the presence of various root canal inhibitors. *Streptococcus gordonii*, *Enterococcus faecalis*, *Lactobacillus acidophilus* and *Actinomyces naeslundii* were used. Dentine, dentine matrix and dead bacteria were selected as inhibitors. The antimicrobial efficacy of monomers of DMADDM and DMAHDM was tested against suspensions formed by mixtures of four bacterial species in or without the presence of inhibitors. The inhibition results were compared with chlorhexidine (CHX) and sodium hypochlorite (NaOCl). One-way analyses of variance (ANOVA) followed by Tukey's multiple comparison test was performed to determine significant differences. The antibacterial effects of DMADDM and DMAHDM were variably inhibited dentine, dentine matrix and dead bacteria. CHX and NaOCl showed substantivity and they inhibited bacteria present in suspension. The concentration of compound decreased in the dentine block due to constant release. Bacterial colonies on the dentine surface and dentine tubules were significantly inhibited by DMADDM and DMAHDM. The antibacterial effects of DMADDM and DMAHDM could be inactivated by dentine, dentine matrix and dead bacteria, while DMADDM and DMAHDM could inhibit bacteria colonization on the dentine surface and kill bacteria present in dentinal tubules. The antibacterial effects of DMADDM and DMAHDM as free monomers in the presence of inhibitors was evaluated for the first time. They could help inhibit the residual bacteria on the dentine surface and in dentinal tubules that may cause persisting infection. Therefore the novel QAMs showed great potentials as root canal medication.

**Keywords:** Quaternary ammonium salt; dimethylaminododecyl methacrylate; dimethylaminohexadecyl methacrylate; root canal inhibitors; antibacterial effects

## 1. Introduction

The main goal of endodontic treatment is to eradicate pulp and periapical tissue infection and promote periapical tissue healing. However, there still exists bacterial persistence within the root canal system even after cleaning and shaping procedures. Therefore, it is important to eliminate the persisting biofilm inside the root canal system before root canal obturation [1]. A wide variety of antimicrobial, disinfecting or cleaning agents such as sodium hypochlorite (NaOCl), chlorhexidine (CHX), ethylenediaminetetraacetic acid (EDTA), citric acid, hydrogen peroxide, have been developed and investigated. Apart from conventional irrigants and intracanal medicaments [2], other measures such as nanoparticle–based disinfectants [3] and photodynamic therapies [4,5] have also been reported. However, the effectiveness of these antimicrobial agents is compromised in the clinical conditions because of the complexity of the root canal anatomy system that permits the localization of bacteria in the inaccessible areas.

Moreover, the complex infected root canal system contains bacteria, bacterial by-products, tissue fluid, dentine, dentine matrix and remnants of pulp tissue. Studies have shown that the disinfecting or antibacterial agents used in endodontics, such as NaOCl [6], CHX [6,7], or calcium hydroxide [1,4,7], can be inactivated by these dentine constituents, tissue remnants or bacterial-related matters. The inorganic component of dentine is supposed to act as tissue inhibitor and it does not allow antimicrobial compounds to penetrate deep inside dentine, which protects bacteria located deep inside dentinal tubules. In addition, the presence of a smear layer after instrumentation or dentine buffering action hinders the effectiveness of antibacterial irrigants and dressings in disinfecting dentinal tubules [8]. Therefore, it is necessary to verify the efficacy of all newly developed antimicrobial compounds for root canal purpose in presence of various tissue inhibitors.

Quaternary ammonium compounds (QAMs) can be co-polymerized and incorporated to dental materials, such as resin composites, adhesives, intracanal medicaments, root canal sealers or root canal irrigants [9–13]. Thus, they can provide long-term contact-inhibition against bacteria [9]. When coming into contact with the bacterial cell wall, QAMs can disrupt the cell wall integrity and the cell ruptures due to its own internal pressure [10]. This unique antibacterial mechanism made QAMs effective against a wide range of microbes and also minimized the possibility of the development of resistance by microbes. Two recently developed QAM compounds, dimethylaminododecyl methacrylate (DMADDM) and dimethylaminohexadecyl methacrylate (DMAHDM), were proven to be more effective than conventional QAMs [11–17]. The resistance against aging is an added benefit of these novel compounds which makes them remain active for a longer duration and inhibit bacterial colonization. The antimicrobial efficacy of DMADDM or DMAHDM used in endodontic treatment as free monomer, not incorporated with other compounds, has not been tested yet. In addition, there have been no reports on the antibacterial effect of dentine powder, dentine matrix and dead bacteria on the efficacy of antibacterial monomers such as DMADDM and DMAHDM.

Therefore, the objectives of this study were to investigate for the first time: (1) the antibacterial effects of DMADDM and DMAHDM in the presence of dentine powder, dentine matrix and dead bacteria; (2) the residual antimicrobial efficacy of DMADDM and DMAHDM; (3) the antibacterial effects of DMADDM and DMAHDM against bacteria inside dentine tubules.

## 2. Materials and Methods

### 2.1. Synthesis of Antimicrobial Compounds

DMADDM was obtained after the chemical reaction between the tertiary ammonium compound, 2-(dimethyl-amino) ethyl methacrylate (DMAEMA) (Sigma-Aldrich, Saint Louis, MO, USA), and the respective organo-halide by a modified Menschutkin reaction [10]. Briefly, 10 mmol of DMAEMA, 10 mmol of 1-bromododecane (BDD) (Sigma-Aldrich) and 3 g ethanol were mixed together, capped and starred at 70 °C for 24 h in vial. After 24 h, the vial was left open for ethanol to evaporate, leaving a clear viscous liquid of DMADDM behind. DMAHDM was synthesized via a modified Menschutkin reaction

where a tertiary amine group was reacted with an organo-halide. Briefly, 10 mmol of 2-(dimethylamino) ethyl methacrylate (DMAEMA), and 10 mmol of 1-bromohexadecane (BHD) (Sigma-Aldrich, Saint Louis, MO, USA) were combined with 3 g of ethanol in a 20 mL scintillation vial. The vial was stirred at 70 °C for 24 h. After that, the solvent was removed via evaporation, yielding DMAHDM as a clear and viscous liquid. The end products were verified as described previously [10]. NaOCl and CHX (Sigma-Aldrich, Saint Louis, MO, USA) were purchased directly from suppliers and the solutions were diluted in sterilized deionized water for working concentration.

### 2.2. Bacterial Strains and Growth Conditions

Four bacteria species, *Streptococcus gordonii* (*S. gordonii*, ATCC 10558), *Enterococcus faecalis* (*E. faecalis*, ATCC 19433), *Lactobacillus acidophilus* (*L. acidophilus*, ATCC 4356), and *Actinomyces naeslundii* (*A. naeslundii*, ATCC 12104), were selected for our experiment. All bacteria strains were provided by the State Key Laboratory of Oral Diseases (Sichuan University, Chengdu, China). Stock inoculums were grown in brain heart infusion (BHI, Sigma-Aldrich, Saint Louis, MO, USA) broth by incubating anaerobically (90% $N_2$, 5% $CO_2$, 5% $H_2$) overnight at 37 °C. The working inoculums were prepared by adding 10 μL suspensions from stock in fresh BHI broth and incubating overnight anaerobically at 37 °C. The concentration of bacteria in working suspension was adjusted to a concentration of $10^9$ colony forming units (CFU, $OD_{600nm}$ = 0.2)/mL. The CFU/mL count in suspension was determined by incubating 10 μL of suspension on a BHI agar plate after serial dilution and counting colonies after 48 h of anaerobic incubation at 37 °C. The mixed suspension of four bacteria at $10^9$ CFU/mL was prepared for subsequent experiments.

### 2.3. Minimum Inhibitory Concentration (MIC) and Minimum Bactericidal Concentrations (MBC) Determination

MIC refers to the lowest concentration of an antibacterial agent that prevents the visible growth of microorganisms under defined conditions. MBC was determined as the lowest concentration of the antimicrobial agent that killed 99.9% of the initial bacterial population.

MIC and MBC of DMADDM and DMAHDM against mixed suspension of four bacteria were identified by broth microdilution, as described previously [11]. The starting concentrations for DMADDM and DMAHDM and CHX were 200 and 100 μg/mL, respectively. Briefly, 200 μL of each antimicrobial agent was transferred into the first well of 96-well microtiter plates (Corning Incorporated, New York, NY, USA) in triplicate. The serial two-fold dilution was performed with 100 μL of BHI broth. Then, 100 μL of overnight grown bacteria suspension adjusted to $10^9$ CFU/mL was transferred into each well. The well with bacteria and the well with antimicrobial compound were accounted as positive and negative control, respectively. After incubation at 37 °C with 5% $CO_2$ for 48 h, the MIC values were determined by visual examination.

After the MIC determination, aliquots of 10 μL bacterial suspensions from the wells with no visible cells, where bacterial growth was inhibited, was seeded in BHI agar plates and incubated under anaerobic conditions for 48 h at 37 °C with 5% $CO_2$. After calculating the colony forming units, the MBC values were determined [11].

### 2.4. Preparation and Analysis of Inhibitors

#### 2.4.1. Synthesis of Inhibiting Factors

The study was permitted by the ethical committee of West China School of Stomatology, Sichuan University. A total of 20 extracted bovine central incisors were used for this study. Samples prepared for the determination of inhibitory effect included (a) dentine powder; (b) dentine matrix; and (c) heat killed cells from four bacterial suspension. All inhibitors were suspended in phosphate buffered solution (PBS). They were prepared as following:

The dentine powder was prepared from root section 1 mm below cemento-enamel junction (CEJ). The teeth dentine were grounded and crushed into powder with a particle size of 0.2–20 microns, as described previously [18]. The prepared dentine powders were washed and centrifuged (4000 rpm, 2 min) three times before sterilization. Sterility was confirmed by incubating the powder overnight in BHI broth.

The dentine matrix was obtained by demineralizing dentine powder with 0.2 M 17% EDTA (pH 7.5) for 2 weeks at 4 °C. EDTA was replaced with fresh solution on every alternate day. EDTA insoluble remaining were dialyzed and lyophilized (4000 rpm, 2 min) with PBS [19].

The dead bacteria were obtained by autoclaving. Dead bacteria in the sample were confirmed by incubating in BHI broth, as described previously [20].

### 2.4.2. Determining Antibacterial Effect of QAMs in the Presence of Inhibitors

The experiment for the inhibition study was adopted from Morgental et al. with some modifications [6]. Briefly, 950 μL of each antimicrobial compound was mixed with 50 μL of bacteria suspension. Then, 100 μL of solute from suspension was spread on an agar plate after serial dilution at 3 min, 10 min, 8 h and 24 h and incubated anaerobically at 37 °C for 48 h. The first two tubes series during dilution contained 900 μL D/E (Dey/Engley) neutralizing broth to minimize the possibility of residue effect of antimicrobial compounds [21]. This part of experiment was considered as control samples.

The experiment of the inhibition of antimicrobial compounds by inhibitors was adopted from Portenier et al. [20]. Briefly, 50 μL of inhibitors were suspended in PBS and were thoroughly mixed by adding 50 μL of antimicrobial compound in it. The suspension was left undisturbed for 1 h at 37 °C. After 1 h, 50 μL of bacteria suspension was added and mixed thoroughly. Then, 10 μL of suspension was incubated and colony-forming unit count were performed.

### 2.5. Analysis of Residual Action Effect of Antimicrobial Compounds

This part of the experiment was incorporated to study further the inhibitory action of dentine on antimicrobial compounds. As done previously, dentine blocks were selected instead of powder or matrix and were incubated in antimicrobial compounds before being inoculated in the bacterial suspension. The inhibitory action was studied by colony counting after serial dilution of suspension as described previously. The process of experiment is described in detail as following:

### 2.5.1. Samples Preparation

The dentine blocks were prepared and stored as described before [22]. Briefly, the root surface was curetted to remove soft tissue attachments and debris from its surface. After debridement, roots were stored in 0.01% NaOCl solution at 4 °C before uses. All 68 dentine blocks (4 mm × 4 mm × 2 mm) were prepared from roots 1 mm below the CEJ. The pulpal surfaces of the dentine blocks were smoothened with 600-grit silicon carbide abrasive paper. The dentine blocks were treated in 5.25% NaOCl for 3 min, followed by 2 min in 17% EDTA to remove the smear layer formed on the surface, and the blocks were finally sterilized by autoclaving. All sterilized dentine blocks were kept in BHI broth for 24 h to ensure complete sterilization. After the confirmation of sterilization, the blocks were stored in PBS. Four groups of antimicrobial agents were prepared for this experiment: 300 μg/mL DMADDM group, 37.5 μg/mL DMAHDM group, 2.0% CHX group and 5.25% NaOCl group.

### 2.5.2. Residual Activity Test

The residual activity test was performed as described previously by Zhang et al. with some modifications [23] (Figure 1). All 68 dentine blocks (4 mm × 4 mm × 2 mm) were divided into their respective groups. Briefly, 48 blocks were assigned for antimicrobial agents (12 blocks per group), 10 blocks in the positive control group, and 10 in the blank group (BHI broth only). The blocks were placed in a 48-well plate, one block per well. The 1 mL antimicrobial agents were added in each well and left for 10 min. The blocks were removed from the antimicrobial solution and excesses of

compounds were removed with sterile paper before transferring the blocks to a 2 mL eppendorf tube which contained 1 mL of previously prepared bacteria suspension ($10^9$ CFU/mL) in BHI supplemented with 0.2% sucrose. Then, 1 mL of BHI was added to the blank control group. All samples were incubated anaerobically at 37 °C for 48 h. The 100-μL suspensions was incubated on agar plates after serial dilution at 3 h, 6 h, 12 h, 24 h, as described above.

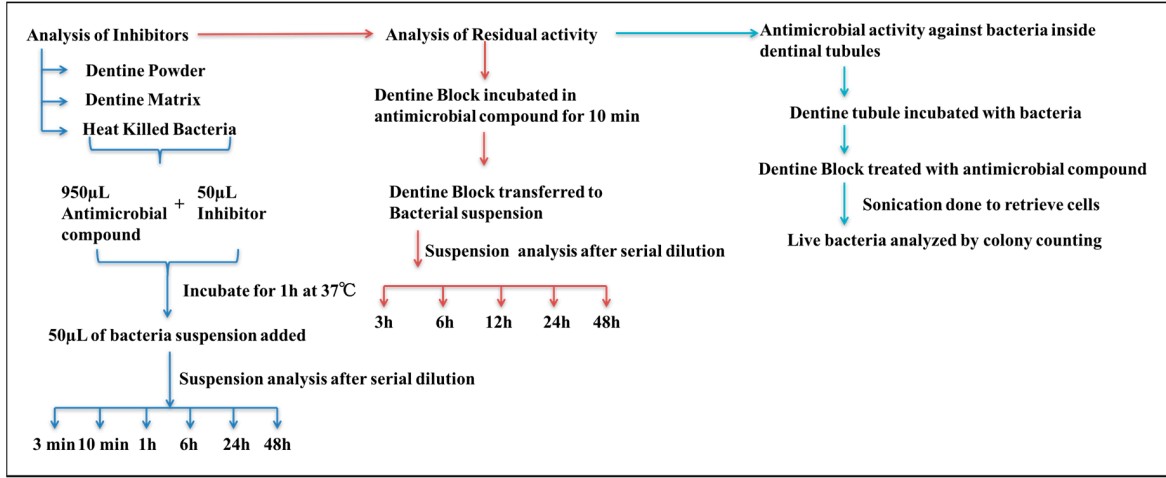

**Figure 1.** Flow-chart of the methods used for the study.

The untreated dentine blocks were inoculated in bacteria suspension for 48 h in BHI supplemented with 0.2% sucrose. After 48 h, half of the dentine blocks from each group were prepared for scanning electron microscope (SEM) analysis and half samples for CFU counts.

Briefly, after 48 h dentine blocks were washed twice with PBS to remove planktonic and loosely adhering bacteria from the surface of the biofilm. The dentine blocks were fixed in 4% glutaraldehyde solution at 4 °C for 4 h. After fixation, dentine blocks were irrigated twice with PBS and dehydrated by immersing in gradually ascending series of ethanol (30%, 50%, 70%, 80%, 85%, 90%, 95%, and 100%) for 15 min. Final dehydration was done with 100% ethanol for 15 min and the dentine blocks were stored in the same solution. Gold sputter coating of the dentine blocks was done before visualizing by SEM (INCA Penta FETX3, OXFORD INSTRUMENT, Abingdon-on-Thames, UK).

Half of the samples were sonicated in a ultrasonic bath (CQ-250A-DST, Hengyue Medical Instruments, Shanghai, China) for 30 min to detach bacteria from the surface. Then, 100 μL of samples were 10-fold serially diluted and 50 μL from several dilutions were spread and incubated on agar plate as described above.

### 2.6. Effects of Antimicrobial Compounds on Bacteria on Dentin Blocks

#### 2.6.1. Samples and Bacteria Inoculation

The preparation and sterilization of dentine blocks were similar to the process described above. For bacteria inoculation, each dentine block was positioned in a microfiltration tube with the pulpal surface of the block facing upwards. The spaces between the tube wall and the dentine block were filled with flowable resin composite (Beautiful flow plus, Shofu, Japan) and light cured for 20 s.

The bacteria inoculation procedure was adopted from Ma et al. with some modifications [22]. Briefly, 500 μL of bacteria suspension in PBS was placed in the microfiltration tube containing the dentinal blocks. The tubes were centrifuged for 10 min at 1000, 2000, 3000 and 4000 rpm in a sequence, twice each time. The solution collected after each centrifugation was discarded and fresh bacterial suspension was added before each new centrifugation cycle.

The dentine blocks were removed after centrifugal cycles, the surrounding resin was removed and the blocks were placed into a 48-well microtiter plate containing 1 mL BHI broth supplemented with 0.2% sucrose for 48 h to resuscitate bacteria and form biofilm.

### 2.6.2. Disinfection Procedure

After 48 h, dentine blocks were rinsed with PBS to remove loosely adhering bacteria. The samples were divided into five groups according to the different methods of antibacterial agent treatment: 300 µg/mL DMADDM group [24], 37.5 µg/mL DMAHDM group [24], 2.0% CHX group, 5.25% NaOCl group, and control group with no treatment. Twelve dentine blocks were assigned for each of the treatment groups. The dentine blocks were placed individually in each well of new 48 wells plate. Then, 50 µL of antimicrobial compound was placed on the pulpal surface of dentine block and left in contact for 10 min. Then, the dentine blocks were washed twice with PBS and placed in 200 µL of D/E neutralizing broth for 10 min to minimize carryover effect of antimicrobial compounds [21]. After that, the dentine blocks with biofilm were harvested by scraping and sonication for 30 min to detach bacteria from the dentinal tubules. The biofilm suspensions were serially 10-fold diluted, spread onto agar plates and incubated at 37 °C aerobically for 48 h. Then, the number of colonies was counted by a colony counter (Reichert, NY, USA) and used, along with the dilution factor, to calculate the CFU counts.

### 2.7. Statistical Analysis

Statistical analyses were performed using IBM SPSS Statistics for Windows, Version 22.0 (SPSS Inc., Chicago, IL, USA). One-way analyses of variance (ANOVA) were performed to detect the significant effects of the variables. Tukey's multiple comparison tests was used to compare the means of each of the groups. The differences of the means of data were considered significant if the $p < 0.05$.

## 3. Results

### 3.1. MIC and MBC of DMADDM and DMAHDM

The MIC and MBC of DMADDM were 50 µg/mL and 100 µg/mL, respectively. The MIC and MBC of DMAHDM were both 12.5 µg/mL.

### 3.2. Inhibitory Action of Antimicrobial Compounds

The influence of different inhibitors on different antimicrobial agents is shown in Figure 2. The bacteria CFUs/mL from PBS are from references. Regarding the DMADDM group and the DMAHDM group, in the initial 3 min, no significant differences in CFU log reduction after exposure to inhibitors ($p > 0.05$) (Figure 2A,B). After 3 min, the DMADDM group and the DMAHDM group were totally inactivated by the presence of dead bacteria, dentine matrix or dentine powder, compared to the group without exposure to any inhibitor. There was no difference between the three different inhibitors ($p > 0.05$). As for CHX group and NaOCl group, the antibacterial effects were not influenced by the inhibitors after 1 h, almost eliminating all bacteria (Figure 2C,D).

### 3.3. Residual Action of Antimicrobial Compounds

The residual activities of antibacterial agents are shown in Figure 3. Compared with the control group, there was no residual antibacterial effect on the dentine block after exposure to DMADDM or DMAHDM. There were no differences in colony count between the control group and QAMs groups ($p > 0.05$). However, in the CHX group and NaOCl group, no bacterial colonies formed on dentine blocks at different time points.

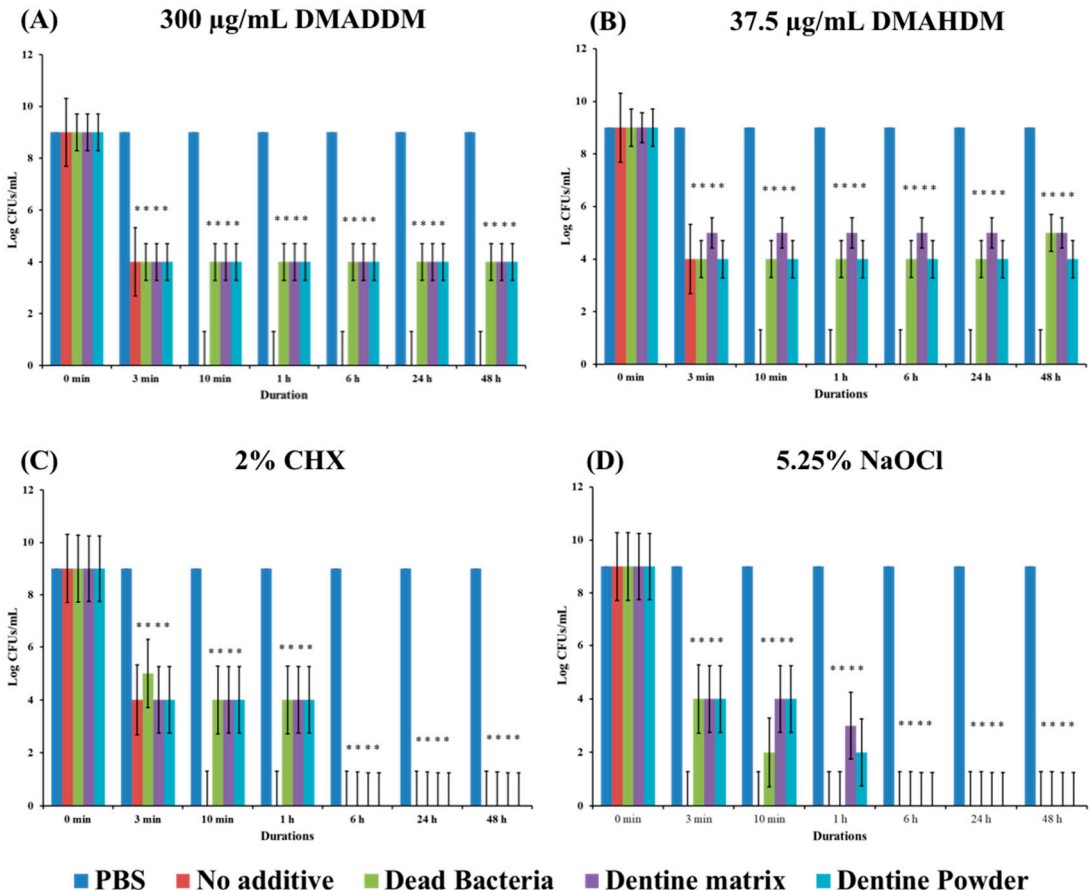

**Figure 2.** Elimination of bacteria from suspension by DMADDM, DMAHDM, CHX and NaOCl in the absence or presence of inhibiting factors. The bar represents the mean value ± standard deviation (n = 9) (* $p < 0.05$).

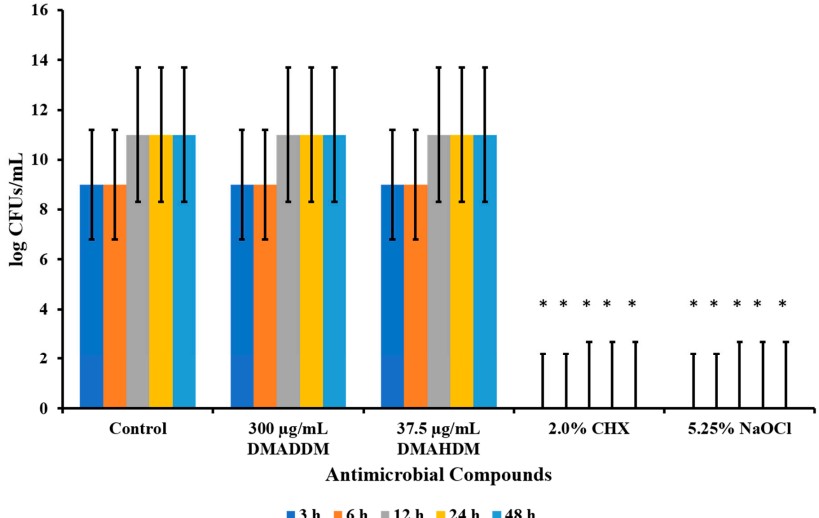

**Figure 3.** The changes in log colony forming units (CFUs)/mL in suspensions from the eppendorf tube containing dentine blocks treated with four antimicrobials compounds (DMADDM, DMAHDM, CHX and NaOCl), and the untreated dentine block (control group). The data is present in mean CFUs ± standard deviation (* $p < 0.05$).

### 3.4. SEM Observation

Figure 4 shows the SEM micrographs of typical 48-h biofilms on dentine disks in different groups. Compared to the dense biofilm of the positive control group (Figure 4A) and the negative control group (Figure 4D), the DMADDM group, DMAHDM group, CHX group and NaOCl group inhibited the development of biofilm in varying degrees. In the treatment groups, few bacteria were seen on the surface of dentine block treated with 300 μg/mL DMADDM (Figure 4B) and 37.5 μg/mL DMAHDM (Figure 4C). Meanwhile, some scattered colonies of bacteria were seen on the surface of the dentine block that was treated with 2.0% CHX (Figure 4E). The dentine block treated with 5.25% NaOCl (Figure 4F) had an eroded dentine surface, reducing lumen of dentinal tubules with few bacteria on the surface.

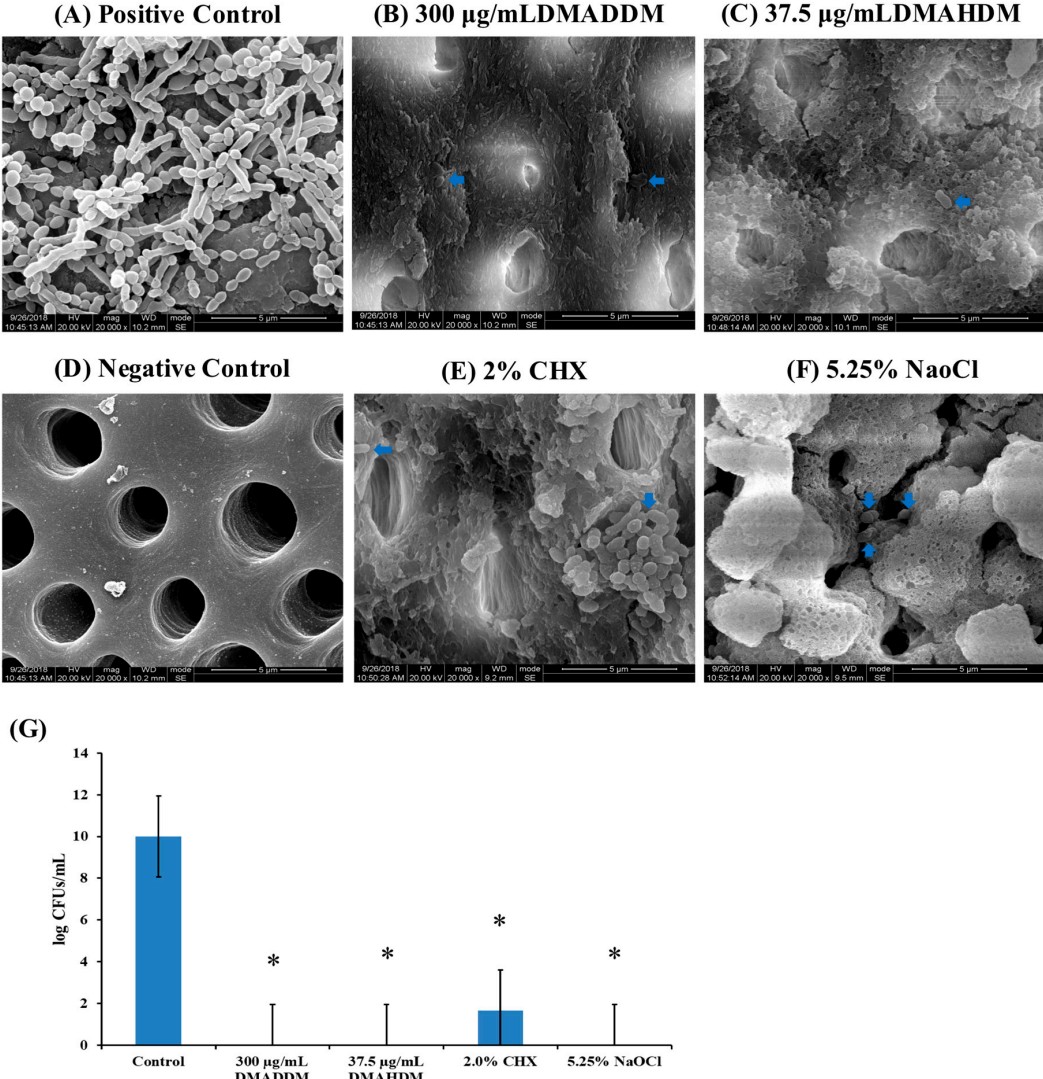

**Figure 4.** Representative SEM micrographs of the typical biofilms in different groups. SEM images of dentine blocks containing no antimicrobial compounds, positive control group (**A**), 300 μg/mL DMADDM group (**B**), 37.5 μg/mL DMAHDM group (**C**), negative control group (**D**), 2.0% CHX group (**E**), and 5.25% NaOCl group (**F**). CFU counts of bacteria in different group are shown in (**G**). Compared to control group, all antimicrobial compounds showed more significant anti-biofilm effects than CHX ($p < 0.05$), with few bacteria on the surface. Among antimicrobial compounds, DMADDM, DMAHDM and NaOCl showed more significant anti-biofilm effects than CHX (* $p < 0.05$). The blue arrow referred to biofilm.

The results of colony count from the dentine block after 48 h of residual activity analysis are shown in Figure 4G. All treated dentine blocks significantly inhibited bacteria colonization on their surface ($p < 0.05$). DMADDM and DMAHDM were equally effective in inhibiting the colonization of bacteria on the surface as NaOCl ($p > 0.05$). Significantly more bacteria were recovered from dentine samples treated with CHX than the other compounds ($p < 0.05$).

### 3.5. Results from Antimicrobial Action on Bacteria Present in Dentinal Tubules

The CFU counts of bacteria inside dentinal tubules after treatment with antimicrobial compounds for 10 min are plotted in Figure 5. The NaOCl group had the strongest antibacterial activity, with about 6 log reductions in CFU ($p < 0.05$). The DMADDM group and the DMAHDM group had CFU results similar to those of the CHX group, equally reducing the CFU counts by about 5 log ($p > 0.05$).

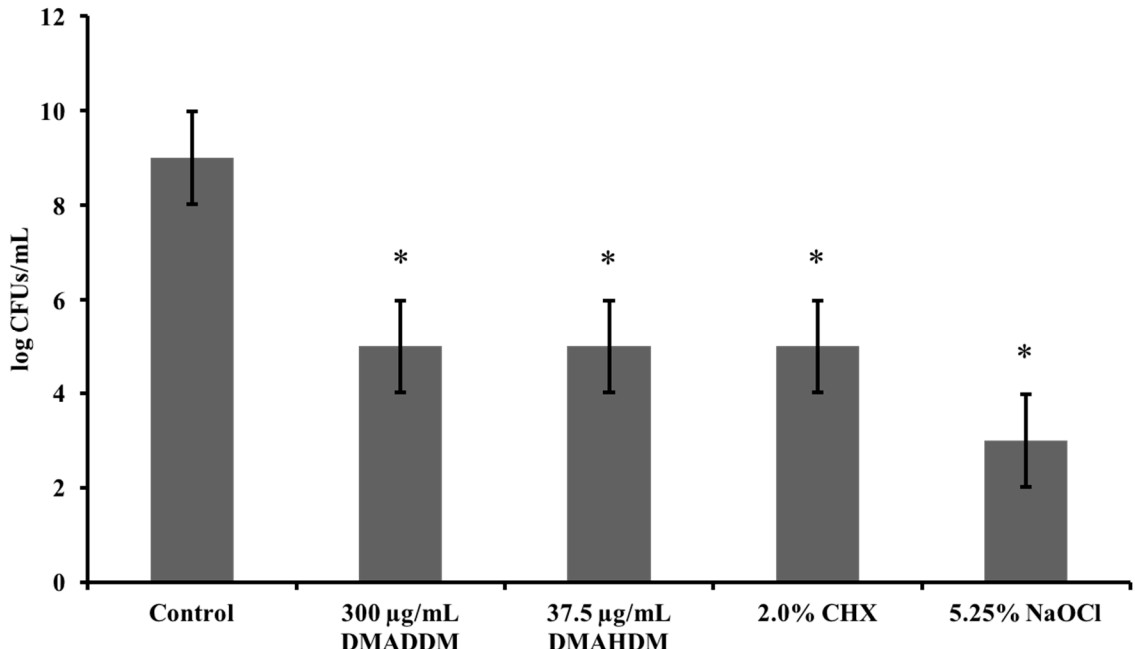

**Figure 5.** CFU counts of bacteria on dentine blocks after treatment with antimicrobial compounds for 10 min. Bacteria were inoculated in dentinal tubules of dentine blocks by centrifugation. The other four groups had fewer CFU counts compared to the control group. The result is presented in mean CFUs ± standard deviation (* $p < 0.05$).

## 4. Discussion

This study examined and compared the inhibition of the antibacterial effect of DMADDM, DMAHDM, CHX and NaOCl by dentine, dentine matrix and dead bacteria. Previous studies have shown that dentine components and dead bacteria possess strong affinity against antimicrobial compounds and reduce their killing efficiency [6,18,20,25]. In the present study, the inhibitory effect of dentine, dentine matrix and dead bacteria against four antimicrobial compounds were detected, with CHX and NaOCl as control group. CHX and NaOCl are the most studied root canal irrigants in endodontics and they were found to be the most effective irrigants against Gram-positive and negative oral pathogens including yeast, because CHX could damage the bacterial cytoplasmic membrane, which subsequent cause the leakage of cytoplasmic material [26]. The antimicrobial effect of NaOCl is based on causing biosynthetic alterations in cellular metabolism and phospholipid destruction, formation of chloramines that interfere in cellular metabolism, oxidative action with irreversible enzymatic inactivation in bacteria, and lipid and fatty acid degradation [27]. The results in the study show that DMADDM and DMAHDM were totally inactivated by the presence of dead bacteria, dentine matrix or dentine powder after 3 min, compared to the group without exposure to any inhibitor.

"Contact killing" was thought to be the antibacterial mechanism of QAMs. In the presence of inhibitors, QAMs could not make good contact with the surface of the bacteria, and the bacteria could survive under the interference of inhibitors. Although CHX and NaOCl seemed to be inhibited by inhibitors before 1 h, after 1 h, the bacteria were eliminated, which was consistent with the previous findings [7]. An explanation for these findings may be that the binding of CHX to dentine modified and prolonged its antibacterial activity against bacteria.

Endodontic-related infection is formed by the symbiotic coaggregation of different species of bacteria [28,29] and multiple species biofilm provides a platform where cooperation between different species of bacteria increases the potential against odds from which all present microbes are benefited [30]. Actinomyces, lactobacillus, streptococcus and enterococcus are commonly isolated microbes from root canals [28,29]. These bacteria play a unique role during biofilm formation. *S. gordonii* and *A. naeslundii* are initial colonizers and *A. naeslundii* provided protection to *S. gordonii* against lethal effect of superoxide [31–33]. *E. faecalis* are frequent isolates from teeth with persisting periapical lesions [34], and their presence increases the survival of other bacteria in the root canal [30]. Interactions between *E. faecalis* and other species found in root canal infections might be important for the development and persistence of periapical disease. Multispecies infection is the main problem for the application of agents under clinical conditions. Therefore, four bacteria species, *S. gordonii*, *E. faecalis*, *L. acidophilus*, and *A. naeslundii* were selected for our study.

This study was aimed at studying the inhibitory effect of dentine components and dead bacteria inside the root canal. The results show that the antimicrobial compounds DMADDM and DMAHDM were inhibited in the presence of dentine powder, dentine matrix and dead bacteria. This finding is similar to previous studies [6,7,18,20,25]. A further experiment was proceeded with to identify the reasons behind inhibition by using dentine blocks and it was speculated that antimicrobial compounds got absorbed and retained by dentine. The retention of the compound was possibly by the dentine matrix because it showed more inhibition of the compound than dentine powder. This finding is confirmed from previous analysis with similar results [25]. After the absorption of compounds by dentine, it is released back into the suspension to inhibit bacteria. This process inhibits the colonization of bacteria on the dentine surface and in the surrounding suspension. DMADDM and DMAHDM did not inhibit bacteria from the suspension but they did not allow bacteria to colonize on the dentine surface. This shows that novel QAMs do not have substantivity but they become permanently incorporated in dentine, maintains their antimicrobial action. This property of novel QAMs will be beneficial against persisting bacteria in root canals and inhibit their propagation to some extent.

Persistent and recurrent periapical infection is due to the presence of bacteria in inaccessible areas such as dentinal tubules [35]. Bacteria present in dentinal tubules are protected from the lethal effects of chemomechanical procedures and baths with antimicrobial compounds [35]. Bacteria inside the dentinal tubule are safe because dentine absorbs antimicrobial compounds and does not allow antimicrobial compounds to pass deeper inside the tubule to kill bacteria. In the present study, DMADDM, DMAHDM, CHX and NaOCl eliminated sufficient bacteria from the dentinal tubule but could not eradicate them totally. NaOCl was the most effective against bacteria inside the dentinal tubule when compared to DMADDM, DMAHDM and CHX. This might be due to its corrosive nature, which removed the superficial layer of dentine and infiltrates deep inside tubule. This process could help disinfect dentinal tubules, but it may compromise the mechanical strength of dentine and leave the tooth susceptible to fracture [36,37]. Therefore, further experiments are needed to display the exact diffusion of the penetration of QAMs through the dentinal tubules to desmodontal structures.

Previous studies have shown the time-dependent inhibition of antimicrobial compounds by inhibitors [6,7,18,20,25]. In the present study, it was found that bacteria were inhibited at 3 min ($p < 0.05$), but no further progress was observed at 10 min in the presence of inhibitory compounds. Considering this, the limitation caused by inhibitors could be overcome with the regular exchange of irrigants and using a large volume. The regular exchange would allow more antimicrobial compounds

to get incorporated into dentine, which could help it to extend its effect deeper inside tubules. Therefore, further study is needed to justify this hypothesis.

CHX has antibacterial substantivity and gets released in the surroundings, which can inhibit bacteria [38]. However, we found that bacteria started to coaggregate as soon as the concentration of CHX decreased on the dentine surface. Contrary to CHX, DMADDM and DMAHDM get incorporated within dentine blocks and remained there to inhibit the aggregation of bacteria on the dentine surface. This shows that dentine bathed with DMADDM and DMAHDM incorporates them but does not inhibit their antibacterial properties on its surface. Our study was limited to 48 h analysis, but the previous study has shown that DMADDM retained its antibacterial effect on dentine surface for 6 months of water aging [39]. On this basis, we can say that dentine bathed with DMADDM can inhibit bacteria for prolonged periods and stop the reoccurrence of endodontic infection. There has been no similar water-aging study done with DMAHDM—since both compounds belongs to same group, we can expect that DMAHDM will also perform in similar manner as DMADDM.

## 5. Conclusions

This study showed that the antibacterial effects of DMADDM and DMAHDM could be inactivated by dentine, dentine matrix and dead bacteria to some extent. However, the antimicrobial compounds DMADDM and DMAHDM could be absorbed in dentine blocks, which helped to inhibit the colonization of bacteria on the dentine surface and kill the bacteria present in dentinal tubules.

**Author Contributions:** Conceptualization, X.Z., M.L. and L.C.; Data curation, S.W.; Formal analysis, Y.H.; Investigation, S.K.T., S.W. and Y.H.; Methodology, S.K.T. and S.W.; Project administration, M.L. and L.C.; Software, S.K.T.; Supervision, X.Z., H.H.K.X., B.R., X.P., M.D.W., M.L. and L.C.; Writing—original draft, S.K.T., S.W. and Y.H.; Writing—review & editing, S.W., Y.H., B.R. and X.P. All authors have read and agreed to the published version of the manuscript.

**Funding:** This study was supported by National Key Research Program of China 2017YFC0840100 and 2017YFC0840107 (L.C); National Natural Science Foundation of China grant 81900993 (S.W) and 81870759 (L.C); International Science and Technology Cooperation Program of Sichuan Province 2017HH0008 (L.C); Innovative Research Team Program of Sichuan Province (L.C); and Key Science and Technology Program of Henan Province 22170123 (X.Y).

**Conflicts of Interest:** The authors declare no conflicts of interest.

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
