# Peer review of "The Antibacterial Effects of Quaternary Ammonium Salts in the Simulated Presence of Inhibitors in Root Canals: A Preliminary In-Vitro Study"

_coatings, doi:10.3390/coatings10020181_

Round 1

Reviewer 1 Report

The following research studies the capabilities of two quaternary ammonium salts,  

dimethylaminododecyl methacrylate (DMADDM) and dimethylaminohexadecyl 17 methacrylate (DMAHDM), to inhibit the growth of four bacterial strains in suspension and the colonization in dentin surface and dentinal tubules. The authors use dentin blocks to detect the bacterial growth on the surface, and microfiltration to detect the bacterial cells in the tubules. As a control the authors used two antimicrobial agents, chlorohexidine (CHX) and sodium hypochlorite (NaOCl).

Please find my following comments:

Pg.2 Lns. 61-63: Please quote a reference.

Pg. 3 Lns. 95-97: L. acidophilus is very getting involved in endodontic infections. 

Figure 1:

Panel B: (a) Should it be DMAHDM instead of DMADDM?

All panels: (a) The asterisk above the data bars of the bacteria incubated for 3 minutes should be re-organized. (b) Why there are no SD values for the data bars obtained from the bacteria group treated with PBS? Is this data being used as a reference? If it does – it should be mentioned in the text. (c) After 1h incubation with the inhibitors, CHX and NaOCl eradicated completely the growth of the bacteria while the QAM compounds continue to eradicate only partially the bacterial growth in the suspension. What is the purpose of showing the data after 1h incubation? (d) The bacterial concentration (control) does not change when the incubation time is increased, which may imply that the bacterial culture is in a non-active stationary phase.   

Figure 2:

The biological purpose of the figure is not clear since there is no residual antibacterial effect on the dentin block after exposure to QAM compound in comparison to the positive controls (CHX and NaOCl). 

Figure 3:

Panel A-E: (a) An image of a negative control sample showing only the dentin block (without treatments and without incubation with bacteria) is required for comparison. (b) What is the purpose of the blue arrow in the images? it is not described in the text.

Pg.7 Lns. 260-261: It is very difficult to observe bacterial cells in the  dentinal tubules from the photos, since we cannot see the surface area of the entire tubule. An accurate measurement is required.

Panel F: (a) what is the meaning of the letters a and b in the graph? (b) photos b and c should be brighter to identify the bacterial cells. (c) in Pg. 7 Lns. 267-268 the authors claim that more bacteria were recovered from dentine samples treated with CHX than other compounds at a significance level of p<0.001 and in the legend of Figure 3F they claim that the antimicrobial compounds showed significantly anti-biofilm effect than CHX at a significance level of p<0.05. Significance levels do not match. (d) I would recommend to re-analyze the significance levels of the differences between the concentration of bacteria recovered from samples treated with CHX and the other treated groups due to the high value of SD in the CHX treated group. (e) Asterisk showing the significance levels should be added to the graph as described in the previous graphs.

Figure 4:

Panel 1: (a) What is the meaning of the letters a, b and c in the graph? (b) How can the authors differentiate between the portion of bacterial cells colonized the dentin block and those colonized the dentinal tubules? The authors did not prove that centrifugation facilitates bacterial cells to colonize only in the tubules. A photo demonstrating that only the tubules are colonized by bacteria should be described. Determining the general CFU counts of bacteria on dentin blocks is more accurate. (c) Asterisk showing the significance levels should be added to the graph as described previously.   

Author Response

Thanks for reviewers’ comments. They are very important for our manuscript and future research work. We have studied the comments carefully, and have made corresponding modification. Here are our responses point by point:

The following research studies the capabilities of two quaternary ammonium salts, dimethylaminododecyl methacrylate (DMADDM) and dimethylaminohexadecyl 17 methacrylate (DMAHDM), to inhibit the growth of four bacterial strains in suspension and the colonization in dentin surface and dentinal tubules. The authors use dentin blocks to detect the bacterial growth on the surface, and microfiltration to detect the bacterial cells in the tubules. As a control the authors used two antimicrobial agents, chlorohexidine (CHX) and sodium hypochlorite (NaOCl).

Please find my following comments:

2 Lns. 61-63: Please quote a reference.

Response: Thanks for your kind suggestions. They have been added in the text as following:

Cheng, L.; Zhang, K.; Melo, M.A.; Weir, M.D.; Zhou, X.; Xu, H.H. Anti-biofilm dentin primer with quaternary ammonium and silver nanoparticles. J. Dent. Res. 2012, 91, 598-604. Antonucci, J.M.; Zeiger, D.N.; Tang, K.; Lin-Gibson, S.; Fowler, B.O.; Lin, N.J. Synthesis and characterization of dimethacrylates containing quaternary ammonium functionalities for dental applications. Dent. Mater. 2012, 28, 219-228. Ge, Y.; Ren, B.; Zhou, X.; Xu, H.H.K.; Wang, S.; Li, M.; Weir, M.D.; Feng, M.; Cheng, L. Novel Dental Adhesive with Biofilm-Regulating and Remineralization Capabilities. Materials (Basel), 2017,10. Feng, J.; Cheng, L.; Zhou, X.; Xu, H.H.; Weir, M.D.; Meyer, M.; Maurer, H.; Li, Q.; Hannig, M.; Rupf, S. In situ antibiofilm effect of glass-ionomer cement containing dimethylaminododecyl methacrylate. Dent. Mater. 2015, 31, 992-1002. Wang, S.; Zhang, K.; Zhou, X.; Xu, N.; Xu, H.H.; Weir, M.D.; Ge, Y.; Wang, S.; Li, M.; Li, Y.; Xu, X.; Cheng, L. Antibacterial effect of dental adhesive containing dimethylaminododecyl methacrylate on the development of Streptococcus mutans biofilm. Int. J. Mol. Sci. 2014, 15, 12791-12806.

3 Lns. 95-97L. acidophilus is very getting involved in endodontic infections. 

Response: Thanks for your kind comments. Some microorganisms are resistant to antimicrobial treatment and can survive in the root canal even after biomechanical preparation including L. acidophilus.

(Narayanan, L.L.; Vaishnavi, C. Endodontic microbiology. J. Conserv. Dent. 2010, 14, 233-239.

Henriques, L.C.F.; Brito, L.C.N.; Tavares, W.L.F.; Teles, W.L.F.; Vieira, L.Q.; Teles, F.R.F.; Ribeiro Sobrinho, A.P. Microbial ecosystem nanlysis in root canal infections refractory to endodontic treatment. J. Endod. 2016, 42, 1239-1245.)

Figure 1:

Panel B: (a) Should it be DMAHDM instead of DMADDM?

Response: Thanks for your careful check. They have been revised to DMAHDM in the text.

All panels: (a) The asterisk above the data bars of the bacteria incubated for 3 minutes should be re-organized.

Response: Yes, they have been revised. Thanks.

(b) Why there are no SD values for the data bars obtained from the bacteria group treated with PBS? Is this data being used as a reference? If it does – it should be mentioned in the text.

Response: Thank you for your valuable comment. The bacteria CFUs/mL from PBS are from the following reference. Sorry, this word got missed from manuscript and we have added this on revised manuscript.

(Tiwari, S.K.; Guo, X.; Huang, Y.N.; Zhou, X.D.; Xu, H.H.; Ren, B.; Peng, X.; Weir, M.D.; Li, M.Y.; Cheng, L. The inhibitory effect of quaternary ammonium salt on bacteria in root canal. Sci. Rep. 2019. 9, 12463.)

(c) After 1h incubation with the inhibitors, CHX and NaOCl eradicated completely the growth of the bacteria while the QAM compounds continue to eradicate only partially the bacterial growth in the suspension. What is the purpose of showing the data after 1h incubation? 

Response: Thank you for your valuable comment. The extended analysis after 1h was to study if antimicrobial compound gets released back to suspension from bonded surface and if it will further inhibit bacteria present in suspension. 

(d) The bacterial concentration (control) does not change when the incubation time is increased, which may imply that the bacterial culture is in a non-active stationary phase.   

Response: Thank you for your valuable comment. The initial 6h analysis showed the increase in bacteria CFUs/mL in suspension. Since the medium was not replaced during analysis and afterward no change in CFUs/mL was observed. The possible reason could be that after 6h bacteria has consumed the available nutrients from media to grow and later on due to deficiency of nutrient and increase in concentration of metabolic waste in medium could have lead bacteria to go for non-active stationary phase.

Figure 2:

The biological purpose of the figure is not clear since there is no residual antibacterial effect on the dentin block after exposure to QAM compound in comparison to the positive controls (CHX and NaOCl). 

Response: Thanks for your kind comments. In our previous studies, QAMs was incorporated into root canal filing paste and they were proved that they could display good antibacterial effect, while the results in the study showed that the monomer DMADDM and DMAHDM have inferior antibacterial effects compared to NaOCl and CHX, which suggested that the free or incorporation condition could affect the antibacterial ability of QAMs. In our next experiment, we will pay more attention to the problem. The newly found problem faced with QAM will also help provide some guidelines for designing or constructing new antibacterial materials. Combined with our previous studies, it was implied that QAMs should be added into root filling sealers to avoid the problems we found in the present study.

Figure 3:

Panel A-E: (a) An image of a negative control sample showing only the dentin block (without treatments and without incubation with bacteria) is required for comparison.

Response: Thanks for your kind comments. Yes, the negative control group has been added into Figure 3.

(b) What is the purpose of the blue arrow in the images? it is not described in the text.

Response: Thanks for your careful comments. The blue arrow referred to biofilm. And the description has been added in the figure caption.

7 Lns. 260-261:It is very difficult to observe bacterial cells in the dentinal tubules from the photos, since we cannot see the surface area of the entire tubule. An accurate measurement is required.

Response: Thanks for your kind comments and suggestions. The residual activity test was performed according to previous study as following. The bacterial cells colonized in the tubules could clearly observed by SEM, which was also reported by previous study as following.

(Zhang, R.; Chen, M.; Lu, Y.; Guo, X.; Qiao, F.; Wu, L. Antibacterial and residual antimicrobial activities against Enterococcus faecalis biofilm: A comparison between EDTA, chlorhexidine, cetrimide, MTAD and QMix. Sci. Rep. 2015, 5, 12944.

Al-Nazhan, S.; Al-Sulaiman, Alaa.; Al-Rasheed, F.; Alnajjar, F.; Al-Abdulwahab, B.; Al-Badah, A. Microorganism penetration in dentinal tubules of instrumented and retreated root canal walls. In vitro SEM study. Restor. Dent. Endod. 2014, 39, 258-264.)

Panel F:(a) what is the meaning of the letters a and b in the graph?

Response: Thanks for your kind comments. Values with dissimilar letters are significantly different from each other. But as you suggested, asterisk showing the significance levels has been added to the graph as described in the previous graphs.

(b) photos b and c should be brighter to identify the bacterial cells.

Response: Yes, thanks for your kind comments. The figures have been revised.

(c) in Pg. 7 Lns. 267-268 the authors claim that more bacteria were recovered from dentine samples treated with CHX than other compounds at a significance level of p<0.001 and in the legend of Figure 3F they claim that the antimicrobial compounds showed significantly anti-biofilm effect than CHX at a significance level of p<0.05. Significance levels do not match.

Response: Sorry about the mistakes. They have been revised as p<0.05.

(d) I would recommend to re-analyze the significance levels of the differences between the concentration of bacteria recovered from samples treated with CHX and the other treated groups due to the high value of SD in the CHX treated group.

Response: Yes, thanks for your kind suggestions. The significance levels of the differences have been re-analyzed.

(e) Asterisk showing the significance levels should be added to the graph as described in the previous graphs.

Response: Thanks for your useful suggestions. They have been revised in the figure.

Figure 4:

Panel 1: (a) What is the meaning of the letters a, b and c in the graph?

Response: Thanks for your kind comments. Values with different letters are significantly different from each other. But as you suggested, asterisk showing the significance levels has been added to the graph as described in the previous graphs.

(b) How can the authors differentiate between the portion of bacterial cells colonized the dentin block and those colonized the dentinal tubules? The authors did not prove that centrifugation facilitates bacterial cells to colonize only in the tubules. A photo demonstrating that only the tubules are colonized by bacteria should be described. Determining the general CFU counts of bacteria on dentin blocks is more accurate.

Response: Thanks for your kind comments and suggestions. The residual activity test was performed according to previous study as following. The bacterial cells colonized in the tubules could clearly observed by SEM, which was also reported by previous study as following.

(Zhang, R.; Chen, M.; Lu, Y.; Guo, X.; Qiao, F.; Wu, L. Antibacterial and residual antimicrobial activities against Enterococcus faecalis biofilm: A comparison between EDTA, chlorhexidine, cetrimide, MTAD and QMix. Sci. Rep. 2015, 5, 12944.

Al-Nazhan, S.; Al-Sulaiman, Alaa.; Al-Rasheed, F.; Alnajjar, F.; Al-Abdulwahab, B.; Al-Badah, A. Microorganism penetration in dentinal tubules of instrumented and retreated root canal walls. In vitro SEM study. Restor. Dent. Endod. 2014, 39, 258-264.)

(c) Asterisk showing the significance levels should be added to the graph as described previously.   

Response: Thanks for reviewer’s comments. They have been revised in the figure.

Reviewer 2 Report

Dear Authors,

Thank you very much for submitting your study entitled „The antibacterial effects of quaternary ammonium 2 salts in the presence of inhibitors in root canals”.

The disinfection of the entire root canal system by chemo-mechanical preparation of the root canals is crucial for a successful endodontic treatment. As we know, mechanical preparation reaches only 50-60% of root canal surfaces but guarantees an efficient chemical disinfection. Nevertheless, activation is indispensable.

I have some major concerns:

1) When reading the manuscript, it becomes evident that authors do not perform endodontic treatment in daily practice. As an endodontist, I get completely different conclusions out of your study:

a) The commonly used agents are well suited as disinfectants in endodontic treatment. Especially NaOCl is crucial due to its dissolving properties. It seems to be superior to all other irrigation solutions or disinfectants. b) DMADDM and DMAHDM showed a decreased activity when inhibitors (dentine, dead bacteria, dentine powder) were present. c) DMADDM and DMAHDM showed inferior antibacterial effects compared to NaOCl and CHX. e) Additionally, when performing endodontic treatment, endodontists always have an irrigation protocol including several agents to achieve a thorough cleaning. This is done to compensate the lack of effectiveness of a single solution.

- it is well known that NaOCl does not affect smear layer

- it is well known that CHX is inactivated by organic tissues but has a high substantivity

- hence, we always have to use an irrigant for smear layer removal

2) Inside the root canals we always have “inhibitors” because dentine and in many cases dead and vial bacteria are present – you selected only 4 bacterial species. You should point out the limitations due to a multispecies infection under clinical conditions. Highlighting the use of those 4 species does not create a better clinical relevance.

Hence, title should contain “in-vitro”

3) MM: I cannot find a concrete explanation for the choice of the concentrations of DMADDM and DMAHDM – did the concentrations exactly correspond to MIC? You did not clarify the choice.

For DMADDM I found concentrations of 20-40µg/mL being lethal for human fibroblasts. What is about diffusion of penetration of such high concentrations through the dentinal tubules to desmodontal structures? This aspect is not discussed at all. That is a major flaw.

4) Setup: I miss a sample size calculation/power analysis

5) MM: A flow-chart containing procedures of all groups would be helpful – numerous subgroups make readability impossible.

6) MM: The investigation period only extended over 48 hours - what is the reason for this period of time? Intracanal dressing were usually placed for 7-10d.

7) Fig 1: Heading of grapb B is incorrect – it should be DMAHDM instead of DMADDM.

8) Fig 3 shows the surfaces of the dentine dics - A picture along the axis of the tubules would be more meaningful. Hence, you cannot prove your statement of the “deep effects” inside the tubules. Penetration depth was not investigated.

7) Discussion: “This study was aimed to study the inhibitory effect of dentine components and dead bacteria inside root canal”

“Further experiment was proceeded to identify the reasons behind inhibition by using dentine block and it was found that antimicrobial compounds got absorbed and retained by dentine”.

=> How did you prove this statement? To my point of view, it is speculative. How does DMADDM or DAMHDM bind to the dentinal structures? Specify “compounds got adsorbed”.

8) Discussion: ll 326 => the sentence is only partially correct – what is about extraradicular infections?

9) Discussion: basic knowledge concerning the other antibacterial agents is poor – explanation of the effect mechanisms of NaOCl and CHX is poorly described.

10) Finally, a major flaw is the missing smear layer in your setup – usually a chemo-mechanical preparation of an infected root canal is performed. Your setup aimed only at non instrumented areas.

To my point of view this limits the clinical relevance, too.

Summarized, this study has several limitations – the only finding is the inhibition effect of dentine, dentine powder and dead bacteria to QAMs. This finding is not new.

I think this topic is better suited being published in an endodontic journal.

Apart from that, language needs major revisions. A native speaker should revise the manuscript.

Author Response

Thanks for reviewers’ comments. They are very important for our manuscript and future research work. We have studied the comments carefully, and have made corresponding modification. Here are our responses point by point:

Thank you very much for submitting your study entitled “The antibacterial effects of quaternary ammonium salts in the presence of inhibitors in root canals”.

The disinfection of the entire root canal system by chemo-mechanical preparation of the root canals is crucial for a successful endodontic treatment. As we know, mechanical preparation reaches only 50-60% of root canal surfaces but guarantees an efficient chemical disinfection. Nevertheless, activation is indispensable.

I have some major concerns:

1) When reading the manuscript, it becomes evident that authors do not perform endodontic treatment in daily practice. As an endodontist, I get completely different conclusions out of your study:

a) The commonly used agents are well suited as disinfectants in endodontic treatment. Especially NaOCl is crucial due to its dissolving properties. It seems to be superior to all other irrigation solutions or disinfectants.

Response: Thanks for your kind comments. Yes, NaOCl is preferred disinfectants in endodontic treatment nowadays. But the problem of bacterial resistance also cannot be ignored. QAMs have been proved that they had long-term contact-inhibition effects against bacteria and lower risk of drug resistance, when compared to other antibacterial drugs. So it is worthy to explore the potentials of QAMs applied in endodontic treatment.

b) DMADDM and DMAHDM showed a decreased activity when inhibitors (dentine, dead bacteria, dentine powder) were present.

Response: Thanks for your kind comments. The results in the study showed that DMADDM and DMAHDM showed a decreased activity, while the newly found problem faced with QAM will also help provide some guidelines for designing or constructing new antibacterial materials. Combined with our previous studies, it was implied that QAMs should be added into root filling sealers to avoid the problems we found in the present study.

c) DMADDM and DMAHDM showed inferior antibacterial effects compared to NaOCl and CHX.

Response: Thanks for your kind comments. In our previous studies, QAMs was incorporated into root canal filing paste and they were proved that they could display good antibacterial effect, while the results in the study showed that the monomer DMADDM and DMAHDM have inferior antibacterial effects compared to NaOCl and CHX, which suggested that the free or incorporation condition could affect the antibacterial ability of QAMs. Combined with our previous studies, it was implied that QAMs should be added into root filling sealers to avoid the problems we found in the present study

e) Additionally, when performing endodontic treatment, endodontists always have an irrigation protocol including several agents to achieve a thorough cleaning. This is done to compensate the lack of effectiveness of a single solution.

- it is well known that NaOCl does not affect smear layer

- it is well known that CHX is inactivated by organic tissues but has a high substantivity

- hence, we always have to use an irrigant for smear layer removal

Response: Thanks for your kind comments. Potent antimicrobial activity, dissolving of remaining pulp tissues with no systemic hazards, reducing instrument friction during mechanical preparation and availability are among the main requirements for an ideal root canal irrigant. Yes, smear layer removal is important for irrigant evaluation. Here in the text, we paid attention to the antibacterial effect of different agents. In the following experiment, we will evaluate the ability of smear layer removal of different agents.

2) Inside the root canals we always have “inhibitors” because dentine and in many cases dead and vial bacteria are present – you selected only 4 bacterial species. You should point out the limitations due to a multispecies infection under clinical conditions. Highlighting the use of those 4 species does not create a better clinical relevance.

Hence, title should contain “in-vitro”

Response: Thanks for your kind comments. They have been revised in the text, as following: “Multispecies infection is main problem for agent application under clinical conditions.”

3) MM: I cannot find a concrete explanation for the choice of the concentrations of DMADDM and DMAHDM – did the concentrations exactly correspond to MIC? You did not clarify the choice.

Response: Thank you for your valuable comment. This experiment is continuation of past experiment published previously titled “The inhibitory effect of quaternary ammonium salt on bacteria in root canal”. The concentrations of DMADDM and DMAHDM were selected as three times the MBC concentration. It has been added into the revised text.

(Tiwari, S.K.; Guo, X.; Huang, Y.N.; Zhou, X.D.; Xu, H.H.; Ren, B.; Peng, X.; Weir, M.D.; Li, M.Y.; Cheng, L. The inhibitory effect of quaternary ammonium salt on bacteria in root canal. Sci. Rep. 2019. 9, 12463.)

For DMADDM I found concentrations of 20-40µg/mL being lethal for human fibroblasts. What is about diffusion of penetration of such high concentrations through the dentinal tubules to desmodontal structures? This aspect is not discussed at all. That is a major flaw.

Response: Thanks for your kind comments. It is worthy to find out the diffusion of penetration of QAMs through the dentinal tubules to desmodontal structures, so it will be studied in our next experiment.

4) Setup: I miss a sample size calculation/power analysis

Response: Thanks for your careful and kind suggestions. They have been added in the text as following: “The teeth dentine were grounded and crushed into powder with a particle size of 0.2-20 microns as described previously [18].” “68 dentine blocks  (4mm ´ 4mm ´ 2mm) were divided in their respective groups.”

5) MM: A flow-chart containing procedures of all groups would be helpful – numerous subgroups make readability impossible.

Response: Thanks for your kind comments. The outline of text has been revised again  to make it clear.

6) MM: The investigation period only extended over 48 hours - what is the reason for this period of time? Intracanal dressing were usually placed for 7-10d.

Response: Thanks for your kind comments. Yes, I agree. In our next experiment, the observation time period will be 7-10d.

7) Fig 1: Heading of grapb B is incorrect – it should be DMAHDM instead of DMADDM.

Response: Thanks for your kind comments. They have been revised in the text.

8) Fig 3 shows the surfaces of the dentine dics - A picture along the axis of the tubules would be more meaningful. Hence, you cannot prove your statement of the “deep effects” inside the tubules. Penetration depth was not investigated.

Response: Thanks for your kind comments. They have been changed to display the antibacterial effect on the surface of dentine dics instead of dentine tubules.

7) Discussion: “This study was aimed to study the inhibitory effect of dentine components and dead bacteria inside root canal”

“Further experiment was proceeded to identify the reasons behind inhibition by using dentine block and it was found that antimicrobial compounds got absorbed and retained by dentine”.

=> How did you prove this statement? To my point of view, it is speculative. How does DMADDM or DAMHDM bind to the dentinal structures? Specify “compounds got adsorbed”.

Response: Thanks for your kind comments. They have been revised in the text. The inappropriate description has been changed to “it was speculated that antimicrobial compounds got absorbed and retained by dentine.”

8) Discussion: ll 326 => the sentence is only partially correct – what is about extraradicular infections?

Response: Thanks for your kind comments. They have been changed to “This property of novel QAMs will be beneficial against persisting bacteria in root canal and inhibit their propagation to some extent.”

9) Discussion: basic knowledge concerning the other antibacterial agents is poor – explanation of the effect mechanisms of NaOCl and CHX is poorly described.

Response: Thanks for your kind comments. They have been revised in the text, as following: “CHX and NaOCl are the most studied root canal irrigants in endodontics and they were found to be the most effective irrigants against Gram-positive and negative oral pathogens including yeast, because CHX could damage the bacterial cytoplasmic membrane, which subsequent cause the leakage of cytoplasmic material [26]. The antimicrobial effect of NaOCl is based on causing biosynthetic alterations in cellular metabolism and phospholipid destruction, formation of chloramines that interfere in cellular metabolism, oxidative action with irreversible enzymatic inactivation in bacteria, and lipid and fatty acid degradation [27].”

( Cieplik, F.; Jakubovics, N.S.; Buchalla, W.; Maisch, T.; Hellwig, E.; Al-Ahmad, A. Resistance toward chlorhexidine in oral bacteria-Is there cause for concern? Front Microbiol. 2019, 10. 87.

Estrela, C.; Estrela, C.R.; Barbin, E.L.; Spanó, J.C.; Marchesan, M.A.; Pécora, J.D. Mechanism of action of sodium hypochlorite. Braz. Dent. J. 2002, 13, 113-117.)

10) Finally, a major flaw is the missing smear layer in your setup – usually a chemo-mechanical preparation of an infected root canal is performed. Your setup aimed only at non instrumented areas.

Response: Thanks for your kind comments. Potent antimicrobial activity, dissolving of remaining pulp tissues with no systemic hazards, reducing instrument friction during mechanical preparation and availability are among the main requirements for an ideal root canal irrigant. Yes, smear layer removal is important for irrigant evaluation. Here in the text, we paid attention to the antibacterial effect of different agents. In the following experiment, we will evaluate the ability of smear layer removal of different agents.

To my point of view this limits the clinical relevance, too.

Summarized, this study has several limitations – the only finding is the inhibition effect of dentine, dentine powder and dead bacteria to QAMs. This finding is not new.

Response: Thanks for your kind comments. The study firstly evaluated the antibacterial effect of agents in vitro because of the multispecies infection under clinical conditions, our next experiment will pay more attention to the QAMs application in vivo. And as shown in our previous studies, the dental materials incorporated with QAMs could play good antibacterial effects, which is also the future application direction.

I think this topic is better suited being published in an endodontic journal.

Apart from that, language needs major revisions. A native speaker should revise the manuscript.

Response: Thanks for your kind comments. We have revised the entire text according to your useful suggestions.

Reviewer 3 Report

In general, this is a correct manuscript. However, it does not contribute anything new to this field of study. Reading the manuscript is a bit complicated and can lead to confusion. There are some drawbacks especially with the methodology used that is simple that hinders the proper evaluation of the antimicrobial activity of the four compounds. The manuscript contains four figures (seven bar charts and five SEM micrographs), and thirty-six references.

There are five keywords proposed. For keywords, where possible, please use Medical Subject Headings terms (MeSH Terms). Some alternative MeSH terms proposed could be “quaternary ammonium compounds”, “dimethylamines”, “methacrylates” or “anti-bacterial agents”.

Throughout the text, when you name bacteria, do it in italics.

In sample preparation, what is the reason for using concentrations of 300 µg/mL of DMADDM and 37.5 µg/mL of DMAHDM? Why did you establish these specific concentrations and no others?

Page 5, line 216. Please indicate which is the neutralizing broth solution.

Page 5, line 223. The correct citation of version 22.0 of the SPSS program is: "IBM SPSS Statistics for Windows, Version 22.0." In the text, you refer to previous versions of the SPSS program.

Forty-eight hours seems insufficient time to properly assess the residual activity of a compound. This is especially striking in the case of NaOCl where its activity in root canal can be extended up to 5 days later as Baca et al.(2011) highlight. The support reference is:

Baca P, Mendoza-Llamas ML, Arias-Moliz MT, González-Rodríguez MP, Ferrer-Luque CM. Residual effectiveness of final irrigation regimens on Enteroccus faecalis-infected root canals. J Endod. 2011;37(8):1121-3.

In the text, you point out that there are no bacteria inside the dentinal tubules. However, the images in Figure 3 do not show the inside of the dentinal tubules, only the presence of bacteria on the dentine surface. The inclusion of images of intratubular infection are needed to support your assertion in the text. Also, with SEM, bacterial viability cannot be verified. 

The discussion is difficult to read and presents disjointed ideas. The study conclusion is very general. Please consider rewriting it, in line with the results of your study.

Author Response

Thanks for reviewers’ comments. They are very important for our manuscript and future research work. We have studied the comments carefully, and have made corresponding modification. Here are our responses point by point:

1): In general, this is a correct manuscript. However, it does not contribute anything new to this field of study. Reading the manuscript is a bit complicated and can lead to confusion. There are some drawbacks especially with the methodology used that is simple that hinders the proper evaluation of the antimicrobial activity of the four compounds. The manuscript contains four figures (seven bar charts and five SEM micrographs), and thirty-six references.

There are five keywords proposed. For keywords, where possible, please use Medical Subject Headings terms (MeSH Terms). Some alternative MeSH terms proposed could be “quaternary ammonium compounds”, “dimethylamines”, “methacrylates” or “anti-bacterial agents”.

Response: Thanks for your kind comments. They have been revised in the text.

2): Throughout the text, when you name bacteria, do it in italics.

Response: Thanks for reviewer’s comments. They have been revised in the text in red.

3): In sample preparation, what is the reason for using concentrations of 300 µg/mL of DMADDM and 37.5 µg/mL of DMAHDM? Why did you establish these specific concentrations and no others?

Response: Thank you for your valuable comment. This experiment is continuation of past experiment published previously titled “The inhibitory effect of quaternary ammonium salt on bacteria in root canal”. The concentrations of DMADDM and DMAHDM were selected as three times the MBC concentration. It has been added into the revised text.

(Tiwari, S.K.; Guo, X.; Huang, Y.N.; Zhou, X.D.; Xu, H.H.; Ren, B.; Peng, X.; Weir, M.D.; Li, M.Y.; Cheng, L. The inhibitory effect of quaternary ammonium salt on bacteria in root canal. Sci. Rep. 2019. 9, 12463.)

4): Page 5, line 216. Please indicate which is the neutralizing broth solution.

Response: Thanks for your kind comments. D/E (Dey/Engley) neutralizing medium was used to neutralizing the carry over effect of antimicrobial compounds.

5): Page 5, line 223. The correct citation of version 22.0 of the SPSS program is: "IBM SPSS Statistics for Windows, Version 22.0." In the text, you refer to previous versions of the SPSS program.

Response: Thanks for reviewer’s careful check. They have been revised in the text as “IBM SPSS Statistics for Windows, Version 22.0.”

6): Forty-eight hours seems insufficient time to properly assess the residual activity of a compound. This is especially striking in the case of NaOCl where its activity in root canal can be extended up to 5 days later as Baca et al. (2011) highlight. The support reference is:

Baca P, Mendoza-Llamas ML, Arias-Moliz MT, González-Rodríguez MP, Ferrer-Luque CM. Residual effectiveness of final irrigation regimens on Enteroccus faecalis-infected root canals. J Endod. 2011;37(8):1121-3.

Response: Thanks for reviewer’s comments. Yes, I agree. In our next experiment, the observation time period will be 7-10d as you suggested.

7): In the text, you point out that there are no bacteria inside the dentinal tubules. However, the images in Figure 3 do not show the inside of the dentinal tubules, only the presence of bacteria on the dentine surface. The inclusion of images of intratubular infection are needed to support your assertion in the text. Also, with SEM, bacterial viability cannot be verified. 

Response: Thanks for your kind comments and suggestions. The negative control group figure has been added into Figure 3. The residual activity test was performed according to previous study as following. The bacterial cells colonized in the tubules could clearly observed by SEM, which was also reported by previous study.

(Zhang, R.; Chen, M.; Lu, Y.; Guo, X.; Qiao, F.; Wu, L. Antibacterial and residual antimicrobial activities against Enterococcus faecalis biofilm: A comparison between EDTA, chlorhexidine, cetrimide, MTAD and QMix. Sci. Rep. 2015, 5, 12944.

Al-Nazhan, S.; Al-Sulaiman, Alaa.; Al-Rasheed, F.; Alnajjar, F.; Al-Abdulwahab, B.; Al-Badah, A. Microorganism penetration in dentinal tubules of instrumented and retreated root canal walls. In vitro SEM study. Restor. Dent. Endod. 2014, 39, 258-264.)

8): The discussion is difficult to read and presents disjointed ideas. The study conclusion is very general. Please consider rewriting it, in line with the results of your study.

Response: Thanks for your kind comments. They have been revised in the text.

Round 2

Reviewer 2 Report

Dear Authors,

Thank you very much for submitting your revised manuscript entitled „The antibacterial effects of quaternary ammonium 2 salts in the presence of inhibitors in root canals in-vitro”.

The manuscript improved by implementation of the revisions, nevertheless the points of criticism concerning the setup and the methodology cannot be changed. Hence, to my point of view, the title should be changed again and should include “preliminary study”.

The investigation interval is not long enough to extrapolate the results to clinical conditions. Intracanal dressings were placed for at least 7-10 days and sealers usually were used for obturation that is aimed to last forever. You always mention (especially in your responses) the potential of the use in sealers but compare it to irrigation solutions. This is inconclusive.

Additionally, it remains unclear if the materials (sealer and other obturation materials) do not change their unique properties when adding quaternary ammonium salts (QAMs)- dimethylaminododecyl methacrylate (DMADDM) and dimethylaminohexadecyl methacrylate (DMAHDM). Probably, the materials are entrapped and cannot achieve their effects.

I strongly believe that your study does not present new findings.

Your conclusion is as follows:

The antibacterial effects of DMADDM and DMAHDM could be inactivated by dentine, dentine matrix and dead bacteria

=> that is true, but you did not prove the activity inside the tubules

While DMADDM and DMAHDM could inhibit bacteria colonization on dentine surface and kill bacteria present in dentinal tubules.

=> it is unproven if the ammonium salts reach the tubules. What is about the penetration depth without dissolution of the smear layer? I cannot follow your conclusion

The antibacterial effects of DMADDM and DMAHDM as free monomers in the presence of inhibitors was evaluated for the first time. They could help inhibit the residual bacteria on the dentine surface and in dentinal tubules that may cause persisting infection. Therefore the novel QAMs showed great potentials as root canal medication.

=> this is speculative. How do they act entrapped in the materials? Are they released by diffusion? 

Summarized, the revision improved the manuscript but failed to convince me and my doubts still predominate.

Author Response

Thanks very much for reviewers’ efforts and kind comments. They are very important for our manuscript and future research work. We have studied the comments carefully, and have made corresponding modification. Here are our responses point by point:

1): Thank you very much for submitting your revised manuscript entitled: “The antibacterial effects of quaternary ammonium salts in the presence of inhibitors in root canals in-vitro”.

The manuscript improved by implementation of the revisions, nevertheless the points of criticism concerning the setup and the methodology cannot be changed. Hence, to my point of view, the title should be changed again and should include “preliminary study”.

Response: Thanks for your valuable comments. The title has been revised into “The antibacterial effects of quaternary ammonium salts in the presence of inhibitors in root canals in-vitro preliminary study.”

2): The investigation interval is not long enough to extrapolate the results to clinical conditions. Intracanal dressings were placed for at least 7-10 days and sealers usually were used for obturation that is aimed to last forever. You always mention (especially in your responses) the potential of the use in sealers but compare it to irrigation solutions. This is inconclusive.

Response: Thank you for your kind comments. A EndoREZ sealer material containing DMADDM has been studied and showed great antibacterial ability in our previous study. The patent about the antibacterial EndoREZ sealer has also been issued (ZL 201610981014.5). The related animal experiment has been done and the results have been submitted in the state of under review.

    At the same time, the antibacterial and anti-biofilm property of DMADDM and DMAHDM against CHX and NaOCl in multispecies biofilm have also been investigated in our previous study. So in the present study, we expected to explore whether the antibacterial effects of DMADDM and DMAHDM could be inactivated by dentine, dentine matrix and dead bacteria to some extent.

(Tiwari, S.K.; Guo, X.; Huang, Y.N.; Zhou, X.D.; Xu, H.H.; Ren, B.; Peng, X.; Weir, M.D.; Li, M.Y.; Cheng, L. The inhibitory effect of quaternary ammonium salt on bacteria in root canal. Sci. Rep. 2019. 9, 12463.

Liu. D.; Peng, X.; Wang, S.P.; Han, Q.; Li, B.L.; Zhou, X.X.; Ren, B.; Xu, H.H.K.; Weir, M.D.; Li, M.Y.; Zhou, X.D.; Cheng, L. A novel antibacterial resin-based root canal sealer modified by Dimethylaminododecyl Methacrylate. Sci. Rep. 2019. 9, 10632.)

3): Additionally, it remains unclear if the materials (sealer and other obturation materials) do not change their unique properties when adding quaternary ammonium salts (QAMs)- dimethylaminododecyl methacrylate (DMADDM) and dimethylaminohexadecyl methacrylate (DMAHDM). Probably, the materials are entrapped and cannot achieve their effects.

Response: Thanks for your kind comments. In our published paper, DMADDM has been added into EndoREZ (the second generation of a methacrylate resin-based sealer). The modified EndoREZ incorporated with DMADDM (1.25% or 2.5%) showed great improved long-term antibacterial ability without compromising the biocompatibility, apical sealing ability and solubility of the sealers. The patent about the antibacterial EndoREZ sealer has also been issued (ZL 201610981014.5). The related animal experiment has been done and the results have been submitted in the state of under review.

(Liu. D.; Peng, X.; Wang, S.P.; Han, Q.; Li, B.L.; Zhou, X.X.; Ren, B.; Xu, H.H.K.; Weir, M.D.; Li, M.Y.; Zhou, X.D.; Cheng, L. A novel antibacterial resin-based root canal sealer modified by Dimethylaminododecyl Methacrylate. Sci. Rep. 2019. 9, 10632.)

4): I strongly believe that your study does not present new findings.

Your conclusion is as follows:

The antibacterial effects of DMADDM and DMAHDM could be inactivated by dentine, dentine matrix and dead bacteria

=> that is true, but you did not prove the activity inside the tubules

Response: Thanks for your kind comments. Yes, the exact antibacterial activity of QAMs inside the tubules is very worthy of study, and in our next experiment, we will pay more attention on it.

5): While DMADDM and DMAHDM could inhibit bacteria colonization on dentine surface and kill bacteria present in dentinal tubules.

=> it is unproven if the ammonium salts reach the tubules. What is about the penetration depth without dissolution of the smear layer? I cannot follow your conclusion

Response: Thanks for your kind comments. The smear layer has been removed by EDTA, as showed in our study “Dentine blocks were treated in 5.25% NaOCl for 3 min, followed by 2 min in 17% EDTA to remove smear layer formed on surface and blocks were finally sterilized by autoclaving.” on Pg. 4 Lns. 166-168. So the quaternary ammonium free monomer solutions could reach the open tubules. But the exact depth of QAMs into the tubules is worthy of our next experiment.

6): The antibacterial effects of DMADDM and DMAHDM as free monomers in the presence of inhibitors was evaluated for the first time. They could help inhibit the residual bacteria on the dentine surface and in dentinal tubules that may cause persisting infection. Therefore the novel QAMs showed great potentials as root canal medication.

=> this is speculative. How do they act entrapped in the materials? Are they released by diffusion? 

Response: Thanks for your kind comments. The present study and our previous study have proved the antibacterial and anti-biofilm property of DMADDM and DMAHDM in multispecies biofilm, which suggested their potentials to be used as endodontic irrigants in future. About the antibacterial effects of QAMs when they were incorporated into the root canal materials, the investigated paper has been published.

    In previous studies, it was shown that antibacterial QAS monomers can copolymerize with other monomers to form polymer matrices that can reduce bacterial growth. Both chemical structures of the main components in EndoREZ (the second generation of a methacrylate resin-based sealer) and DMADDM feature double bonds that may form cross-link structures under certain conditions. So DMADDM could not be released after mixing with EndoREZ, but have a “contact killing” antibacterial mechanism, as QAMs immobilized materials surfaces were highly positively charged, which can attract the negatively charged bacteria, furthermore, the bacteria membrane would be penetrated and interrupted by the long fatty alkyl chains of QAMs. Therefore, the EndoREZ sealer containing DMADDM (1.25% or 2.5%) displayed long-term antibacterial ability without compromising the biocompatibility, apical sealing ability and solubility of the sealers.

(Tiwari, S.K.; Guo, X.; Huang, Y.N.; Zhou, X.D.; Xu, H.H.; Ren, B.; Peng, X.; Weir, M.D.; Li, M.Y.; Cheng, L. The inhibitory effect of quaternary ammonium salt on bacteria in root canal. Sci. Rep. 2019. 9, 12463.

Liu. D.; Peng, X.; Wang, S.P.; Han, Q.; Li, B.L.; Zhou, X.X.; Ren, B.; Xu, H.H.K.; Weir, M.D.; Li, M.Y.; Zhou, X.D.; Cheng, L. A novel antibacterial resin-based root canal sealer modified by Dimethylaminododecyl Methacrylate. Sci. Rep. 2019. 9, 10632.)

7): Summarized, the revision improved the manuscript but failed to convince me and my doubts still predominate.

Response: Thanks for your kind comments. We have revised the entire text according to your useful suggestions.

Reviewer 3 Report

I greatly appreciate the effort of the authors to achieve a clear improvement in their revised manuscript. All my suggestions have been duly addressed.

Author Response

Thanks very much for your efforts and kind comments.

Round 3

Reviewer 2 Report

Dear Authors,

Thank you very much for the revised manuscript. The manuscript improved due to the implemented revisions.

This manuscript seems suited being published, now.

Nevertheless, the title should be revised. The title sounds strange. I think there should be a colon to separate the title.

The antibacterial effects of quaternary ammonium salts in the simulated presence of inhibitors in root canals: a preliminary in-vitro study 

You did not perform a study in root canals - you used dentin blocks. Hence, it is a simulation of the clinical situation - this should also be implemented in the title

Author Response

Thanks very much for reviewers’ efforts and kind comments. They are very important for our manuscript and future research work. We have studied the comments carefully, and have made corresponding modification. Here are our responses point by point:

Thank you very much for the revised manuscript. The manuscript improved due to the implemented revisions.

This manuscript seems suited being published, now.

Nevertheless, the title should be revised. The title sounds strange. I think there should be a colon to separate the title.

The antibacterial effects of quaternary ammonium salts in the simulated presence of inhibitors in root canals: a preliminary in-vitro study 

You did not perform a study in root canals - you used dentin blocks. Hence, it is a simulation of the clinical situation - this should also be implemented in the title

Response: Thanks for your valuable comments. The title has been revised in the text as you suggested.